# Relation between Exogenous Insulin and Cognitive Function in Type 2 Diabetes Mellitus

**DOI:** 10.3390/medicina57090943

**Published:** 2021-09-07

**Authors:** Diana Šimonienė, Džilda Veličkienė

**Affiliations:** 1Department of Endocrinology, Academy of Medicine, Lithuanian University of Health Sciences (LUHS), 44307 Kaunas, Lithuania; dzilda.velickiene@lsmuni.lt; 2Department of Endocrinology, Kaunas Clinics, Hospital of LUHS, 50161 Kaunas, Lithuania

**Keywords:** type 2 diabetes, insulin resistance, mild cognitive impairment

## Abstract

*Background and objectives*: Although the role of insulin in the periphery is well understood, not as much is known about its multifactorial role in the brain. The aim of this study is to determine whether exogenous insulin, evaluated by daily insulin requirement, has an impact on mild cognitive impairment (MCI), and whether this relationship is mediated by insulin doses and other risk factors. *Materials and methods*: A sample of 100 participants with type 2 diabetes aged 40 and over was divided into case and control groups, according to their insulin requirement. Patients with an insulin requirement >1 IU/kg/day were assessed as the case group whereas those with an insulin dose <1 IU/kg were used as the control group. All participants underwent cognitive testing using MoCA questionnaire scoring and blood analysis to determine lipid and uric acid levels in plasma. Subjects were categorized as having normal cognitive function or MCI. *Results*: Results showed that the prevalence of MCI in Lithuanian elderly diabetic patients was high in the groups with a normal insulin requirement or high insulin requirement at 84.8% and 72.5%, respectively (*p* = 0.14). Age (*p* = 0.001) and insulin dose (*p* < 0.0001) were related to the MCI. Using ROC curve analysis, the highest rate risk of MCI occurred when the insulin dose was lower than 144 IU/d. *Conclusions*: In summary, the results of this study provided evidence that increased exogenous insulin supply improves cognitive function. Higher insulin dose (>144 IU/d) demonstrated a positive effect on cognitive function, especially in individuals with poorly controlled diabetes (HbA1c ≥ 9%). Finally, the prevalence of MCI in the T2DM population was found to be very high. Future research is needed to determine whether high exogenous insulin doses have a protective effect on MCI.

## 1. Introduction

Type 2 diabetes mellitus (T2DM) is one of the most prevalent metabolic diseases around the world. The impact of diabetes on microvascular and macrovascular systems has been proven. Insulin resistance in the pathogenesis of T2DM is a mandatory factor. The role of insulin resistance on diabetes complications is also obvious. Diabetes also presents a vascular risk factor for cerebrovascular lesions that may cause cognitive impairment [1]. Compared to people without diabetes, those with diabetes have a greater risk of cognitive decline [2]. Thus, cognitive function impairment is also a known complication of diabetes [3]. An 11-year-long Finnish observational study demonstrated that insulin resistance was an independent predictive factor for cognitive function decline in the healthy adult population [4]. Our understanding of the effect of insulin resistance on cognitive function in the T2DM population is not yet clear.

The exact mechanism of decline in cognitive function in diabetes is not clear. It is thought that T2DM, as well as other metabolic syndromes, causes a central insulin resistance, which in turn leads to the disruption of insulin signaling pathways and as a consequence induces accelerated Aβ deposition and facilitates the polymerization of the *tau* protein, which is a possible mechanism that initiates dementia [5].

Determining insulin resistance in T2DM is not simple, especially in patients who are treated with exogenous insulin because of exogenous hyperinsulinaemia. The gold standard for exact and correct insulin sensitivity evaluation is the hyperinsulinemic–euglycemic clamp technique [6]. However, this method is impractical, as this is a time-consuming and invasive experimental technique. Most studies evaluating insulin resistance use the pancreatic islets function (HOMA-islet) and the Insulin Resistance Index (HOMA-IR) [4,7,8]. Evaluation of insulin requirement is an alternative method for the determination of insulin resistance. However, this method is not worldwide accepted. It is known that patients who require >1 unit/kg/day of insulin are considered to have insulin resistance [9].

There are many tests used to evaluate cognitive function. These tests vary by age, education, social class, and living situation (e.g., living alone at home or in a geriatric institution), as well as by decline of cognitive function and the severity of dementia. The Montreal Cognitive Assessment (MoCA) is one of the most useful screening instruments for the evaluation of cognitive impairment [10,11,12].

There are several studies that have evaluated the impact of T2DM on cognitive function, and most of them demonstrated decrements in the cognitive function in patients with T2DM [1,4,7,8,13]. Most of them evaluated insulin resistance using the HOMA-islet and HOMA-IR indexes [4,7,8]. However, many studies with diabetic patients and intranasal insulin demonstrated the protective effect of exogenous insulin on cognitive function [14,15]. Therefore, the data are controversial. Moreover, the impact of exogenous insulin and insulin resistance on cognitive function is unclear. Insulin resistance where the insulin requirement is high and the glucose levels are maintained by high insulin doses is related to bad glycemic control. Uncontrolled T2DM is related to the progression of diabetes complications, including cognitive functions [16]. Although there is a large body of evidence demonstrating an association between diabetes and cognitive decline or dementia, most studies focus on hyperglycemia and cognition but do not discuss insulin levels or insulin resistance. Moreover, previous data collected in Lithuania have shown a different prevalence of MCI compared to other countries, and therefore, an adjustment is necessary to determine whether it is a random phenomenon or not.

We conducted this study in order to verify the hypothesis that treatment with high exogenous insulin doses that indirect refer insulin resistance in the diabetic population, could have more expressed effect on cognitive function. The main goals of our study are as follows:To evaluate the prevalence of cognitive impairment in T2DM patients with different insulin requirements.To evaluate insulin doses and other factors in relation with cognitive impairment.To assess the impact of exogenous insulin on cognitive function.

## 2. Material and Methods

### 2.1. Patient Selection

Between 2017 and 2020, 100 patients with T2DM, aged 18–80, were recruited by the department of endocrinology of the Lithuanian University of Health Sciences. Written consent was obtained from the participants at the beginning of the study.

### 2.2. Study Design

Case–control study. Patients were assigned to the case group, in which they received a high insulin dose for treatment of T2DM (>1 IU/kg/day) and had poorly controlled diabetes (HbA1c ≥ 9%) (*n* = 52). Patients with relatively low or normal insulin requirements (<1 IU/kg/day) and adequately controlled diabetes (HbA1c < 8%) were assigned to the control group (*n* = 48). Each patient from the control group was matched with a case group subject according to gender, age, and diabetes duration, with an allowed deviation of 2–4 years from those in the case group.

### 2.3. Inclusion and Exclusion Criteria

Inclusion criteria for the case group: diagnosed T2DM, age 18–80 years old, written informed consent, combination therapy with insulin (>1 IU/kg/day) and metformin, except when metformin is contraindicated or not tolerated by the patient, HbA1c ≥ 9%.

Inclusion criteria for the control group: diagnosed T2DM, age 18–80 years old, written informed consent, treatment with insulin (<1 IU/kg/day) with or without oral antidiabetic medicine, HbA1c < 8%.

Exclusion criteria: over 80 years old, severe renal impairment (defined as estimated glomerular filtration rate (eGFR) < 30 mL/min by Modification of Diet in Renal Disease (MDRD) formula or under dialysis), active cardiovascular disease, as well as any other vascularevent (such as myocardial infarction, stroke, acute periphery artery disease within 3 months of prior inclusion in the study OR congestive heart failure New York Heart Association (NYHA) IV requiring hospitalisation within 1 year prior to inclusion in the study, OR history of cardiac arrhythmia that required hospitalization, emergency cardioversion or defibrillation within 3 months prior to inclusion in the study), oncological disease within the last six months before the study, ongoing treatment with glucocorticoids, previously diagnosed dementia.

### 2.4. Methods

#### 2.4.1. Questionnaires, Laboratory Tests

Participants filled out questionnaires about demographic data (such as age, gender, diabetes duration in years, education, marital status, dependence on smoking) and their current insulin dose and hypertension status. Education was divided into two groups: those with basic (lower secondary) education or upper secondary education were referred to as lesser educated. Those who had completed higher university education or non-university higher education were considered as more highly educated patients. Weight and height were measured while participants were wearing light clothing and no shoes. Height was assessed with a 0.1 cm accuracy using a wall stadiometer. Weight was measured with a digital scale with a 0.1 kg accuracy. Obesity was defined as body mass index (BMI) above ≥30 kg/m^2^. We also evaluated parameters of metabolic syndrome, which is inseparable from insulin resistance. According to the National Cholesterol Education Program (NCEP) Adult Treatment Panel III (ATP III), metabolic syndrome is defined as abdominal obesity (waist circumference (102 cm in men or 88 cm in women)), with hypertension (blood pressure >130/85 mmHg) and hyperlipidemia (plasma triglycerides >1.7 mmol/L, HDL cholesterol <1.03 mmol/L in men or <1.29 mmol/L in women) [17]. The most recent value of HbA1c and values of lipids on the day of administering the MoCA test were taken from the electronic medical records. Uric acid, measured at fasting state in the hospital laboratory. The upper limit of normal uric acid assay was 518 µmol/L.

#### 2.4.2. MoCA Questionnaire

We chose to use the MoCA questionnaire for this study because it is a validated cognitive screening tool with high sensitivity (90%) and specificity (87%) for detecting MCI [18]. MoCA is a diagnostic tool for the early and accurate detection of cognitive disorders. The MoCA test was performed for all study subjects by a single trained and certified researcher. The Lithuanian version of the MoCA test with a 30-point score was used.

The MoCA test consists of 7 sub-scores: visuospatial/executive skills (5 points); naming (3 points); memory (5 points for delayed recall); attention (6 points); language (3 points); abstraction (2 points); orientation (6 points). The maximum possible score was 30. One point was added if the subject had less than 12 years of formal education [18]. Patients with MoCA scores ≥26 were considered to have normal cognition and those with MoCA scores <26 as MCI. The upper level of the MoCA score for MCI is 25. Patients with dementia were not excluded from the study, except in cases where severe dementia was already diagnosed, prior to the study. The prevalence of MIC assessed by a MoCA score under 26 in groups with different resistance to insulin was evaluated as well.

### 2.5. Compliance with Ethics Guidelines

All executed procedures followed the rules of good clinical practice and the proper ethical standards and were in accordance with the 1975 Declaration of Helsinki, revised in 2013. The study was approved by the Kaunas Regional Biomedical Research Ethics Committee (BE-2-29, No SRI—01 version 2, 2017-05-17) before the start of the study. Informed consent was obtained from all patients before involvement in the study.

### 2.6. Statistical Methods

Statistical analyses were performed using Statistical Package for Social Sciences ( version 22, SPSS, Inc., Chicago, IL, USA). Frequency and percentage were calculated to estimate the prevalence of cognitive impairment in T2DM. Descriptive statistics were used to summarize all measurements. The Kolmogorov–Smirnov test allowed us to check that the results of the sample corresponded to the normal distribution. Comparisons between the two means of independent samples were analyzed using Student’s *t*-test. For comparison of categorical variables, the χ2 test was performed. Quantitative data were tested with Pearson’s correlation coefficient (r). Comparisons within more than two groups were made using the Kruskal–Wallis test. In order to predict the dependent variables, binary logistic regression was performed. Receiver operating characteristic (ROC) curve analysis was chosen for the evaluation of the highest risk of MCI in diabetic patients treated with insulin. The area under the curve (AUC) was calculated in order to estimate the diagnostic accuracy. The optimal cut-off points were determined according to the Youden criteria (J_max_). The level of statistical significance was set as *p* < 0.05.

## 3. Results

The clinical characteristics of two groups with different insulin requirement were compared (Table 1).

When the MoCA score was ≥26, cognitive function was referred to as normal. It was found that the overall prevalence of impaired cognitive function was ~79% and a significant difference between groups was not observed, since the prevalence of MIC was similar in both groups (*p* = 0.14) (Figure 1).

According to the total MoCA score, there was no statistical difference between groups with different insulin requirements. Thus, further analysis was carried out by assessing the overall distribution of cognitive function in the sample without grouping. Overall relations of cognitive function with various factors (such as age, insulin dose, HbA1c, metabolic syndrome, BMI, metformin usage and diabetes duration) in T2DM were analyzed. Only age and insulin dose demonstrated significant relation with cognitive function (see Figure 2 and Figure 3). It was found that age and insulin doses correlated with the MoCA score: with increasing age, the total MoCA score decreases (r = −0.31, *p* = 0.002) (see Figure 3); on the contrary, a higher insulin dose showed a positive direct correlation with MoCA score (r = 0.27; *p* = 0.008) (see Figure 2).

The relation of the aforementioned factors with MCI (expressed by a MoCA score inferior to 26) was evaluated, and it was found that the insulin dose (*p* = 0.000) was exclusively related to MCI. Among patients with type T2DM and MCI, the mean daily insulin dose was 99.14 ± 48.11 IU/d. Meanwhile, among patients with T2DM and normal cognitive function, the mean insulin dose was 134.90 ± 58.13 IU/d (*p* = 0.005). Moreover, education was correlated with MCI. Participants with lower education tended to have MCI (*p* = 0.08) more often.

The relation of MCI and additional metformin use was evaluated. There were no interactions between metformin use and a better MoCA score (Table 2).

Using ROC curve analysis, we evaluated patients with high and low risk for MCI, based on the risk assessment model. In this sample, the area under the ROC curve (AUC = 0.68, *p* = 0.005) showed that a diabetic patient using an insulin dose lower than 144 IU/d (with 82.9% sensitivity and 52.4% specificity) had a 68% chance of having the highest risk rate of MCI (Figure 4). J_max_ = 0.35.

So, the insulin dose >144 IU/d (from ROC curve) was associated with better cognitive functioning (OR 5.33 (95% CI 1.87–15.14), *p* = 0.002). The results were adjusted for age, but the role of insulin on MCI did not change (OR 5.25 (95% CI 1.83–15.07), *p* = 0.002). Other factors, such education, gender, HbA1c, and diabetes duration did not affect the role of insulin on MCI. If only the results of BMI were adjusted, insulin role on MCI significantly decreased (OR 3.8 (95% CI 1.24–11.60), *p* = 0.02).

## 4. Discussion

The main goal of this study was to determine whether insulin resistance expressed by exogenous insulin doses and its daily requirement had an impact on MCI. According to various studies, the MoCA test appeared to be superior to other tests, due to its high specificity and sensitivity for MCI evaluation [19].

In this study, we found that the prevalence of MCI in Lithuanian elderly diabetic patients with mild or severe insulin resistance was 84.8% and 72.5%, respectively, which is higher than the worldwide literature demonstrates. For example, one Chinese population-based study demonstrated that the incidence of MCI in diabetic patients was around 21.8% [20]. Chinese authors evaluated MCI using Petersen’s MCI diagnostic criteria. In our study, MCI was evaluated according to the final MoCA score (score <26 referred MCI). The same study found that high education was a protective factor of cognition, similarly to our study. Another study reported that the prevalence of MCI using the MoCA questionnaire in elders with T2DM was 31.5% [21] or 28% [22]. Scientists from Poland revealed a prevalence of MCI was 31.5%. Higher HbA1c level, previous CVD, hypertension, retinopathy, increased number of comorbidities, and fewer years of formal education were variables that increased the likelihood of being diagnosed with MCI [21]. Finally, in other worldwide studies, different methods and evaluation scales were used to determine MCI in T2DM populations. We suspect this disparity might be associated with the different applied MCI diagnostic criteria. Therefore, it is very important to unify the diagnostic criteria for MCI.

The reason for high MCI prevalence in our study compared to other studies might be due to the single test used to indicate MCI. Still, one Lithuanian study involving 121 subjects withT2DM found that the prevalence of impaired cognitive function was 79.3% among all subjects [23], which is very similar to our results. High MCI prevalence may be due to Lithuanian dietary habits and differences in lifestyle compared to China or Western countries. Moreover, the vast majority of participants in this study had many comorbidities that may have had a negative impact on the final MoCA score. Moreover, it is obvious that additional studies in the Lithuanian population are needed.

It is known that the hormone insulin has a number of important effects on the central nervous system of a healthy person [7,24]. Numerous studies have proved that patients with diabetes have an increased risk of developing dementia. This relation is associated with insulin deficiency or dysfunction of insulin receptor signaling due to insulin resistance [7,24]. Logically, treatment with exogenous insulin restores the effect of insulin through insulin signaling pathways, and as a consequence, brain function improves, including memory. Furthermore, several studies have demonstrated that interventions or treatments with exogenous insulin alleviated dementia symptoms in patients with insulin resistance and improved their Mini-Mental State Examination (MMSE) scores [14,15]. This study demonstrated that increasing exogenous insulin supply, which indirectly refers to insulin resistance, improves cognitive function in individuals with poorly controlled diabetes (HbA1c ≥ 9%). Thus, either supplying more exogenous insulin overcomes the potential insulin resistance or increasing insulin administration increases cognitive function.

There is a lack of clinical trials evaluating the efficacy of different diabetes treatments and specifically no data on insulin therapy in adults with T2DM and cognitive impairment.

Similar conclusions, namely that high exogenous insulin doses might be an important risk factor for better cognitive function in patients with T2DM, were obtained in a Chinese study, where authors analyzed the HOMA-IR of 212 elderly T2DM subjects. They concluded that insulin resistance (evaluated by HOMA-IR) and higher education were protective factors for cognitive impairment in elderly patients with T2DM [13]. However, in this Chinese study, participants treated only with insulin were not included. Nevertheless, this study adequately set apart T2DM patients with insulin resistance who do not need insulin and only use oral antidiabetic drugs from patients who need insulin and present MCI.

However, it is known that insulin therapy is related to a high risk of hypoglycemia, which in turn leads to the decline in cognitive function [25]. Reports from China, which included 78 patients with T2DM, divided subjects into an MCI group (<26, *n* = 48) and a normal group (≥26, *n* = 30) according to their MoCA score, before examining HOMA-IR and HOMA-islet, and compared the findings between the two groups. The authors concluded that insulin resistance is a risk factor for MCI and can be a biomarker for the prediction of MCI in patients with T2DM [8]. Unfortunately, there was no information about the treatment of the included patients.

In summary, the number of studies examining the relationship between insulin resistance, insulin treatment, and cognitive functions is increasing, as the need for them is high, since the mechanisms and results remain inconsistent.

The present study had some limitations. First and foremost, our results depended on a relatively small sample size and, thus, may not be generalized for the majority of patients. Due to the small sample, no significant differences were found in the groups assessing other risk factors for impaired cognitive function. It is obvious that additional studies with larger samples in the Lithuanian population are needed.

Our study showed confounding biases.

MCI was assessed only via the MCoA fast test, instead of through combined tests, which would have improved the data about cognitive impairments. The high prevalence of MCI compared to other studies might be due to the single test used to indicate MCI. Finally, the insulin resistance expressed by HOMA was not measured in patients, and the presence of endogenous insulin secretion measured by c-peptide was not taken into account, which would have improved the data about insulin resistance and might also have provided interesting insights into insulin sensitivity and cognitive functions.

This study also showed some merit. This is the first study that has assessed insulin resistance according to daily insulin requirements and evaluated the effects of exogenous insulin on cognitive functions. Perhaps this is why some controversial results were received. Moreover, in this study, the cut-off of insulin dose for better cognitive function was evaluated. Undoubtedly, a similarly designed study with a larger sample is needed to confirm or reject the obtained relation.

In summary, the results of this study provide evidence that increased exogenous insulin supply improves cognitive function. Higher insulin dose (>144 IU/d) demonstrated a positive effect on cognitive function, especially in individuals with poorly controlled diabetes (HbA1c ≥ 9%). Finally, the prevalence of MCI in the T2DM population is very high.

## Figures and Tables

**Figure 1 medicina-57-00943-f001:**
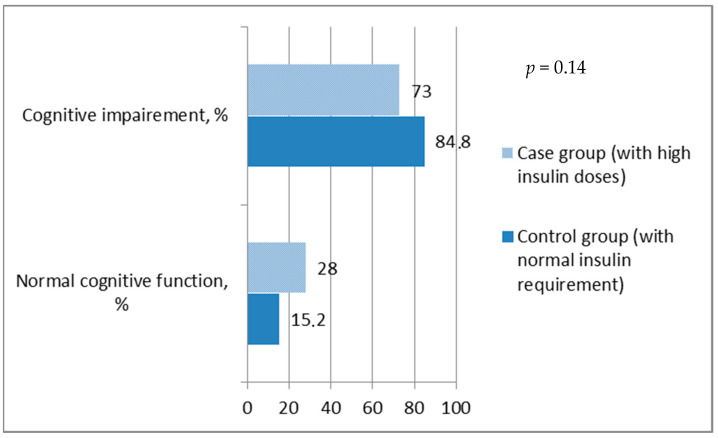
Prevalence of impaired cognitive function in T2DM groups with different exogenous insulin doses.

**Figure 2 medicina-57-00943-f002:**
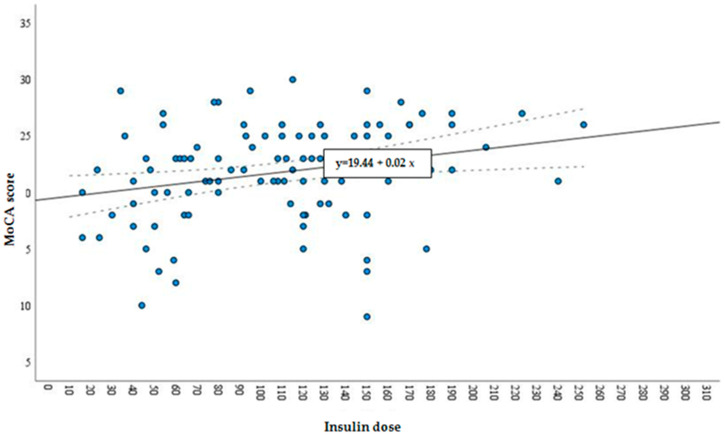
Linear correlation of insulin dose with MoCA score.

**Figure 3 medicina-57-00943-f003:**
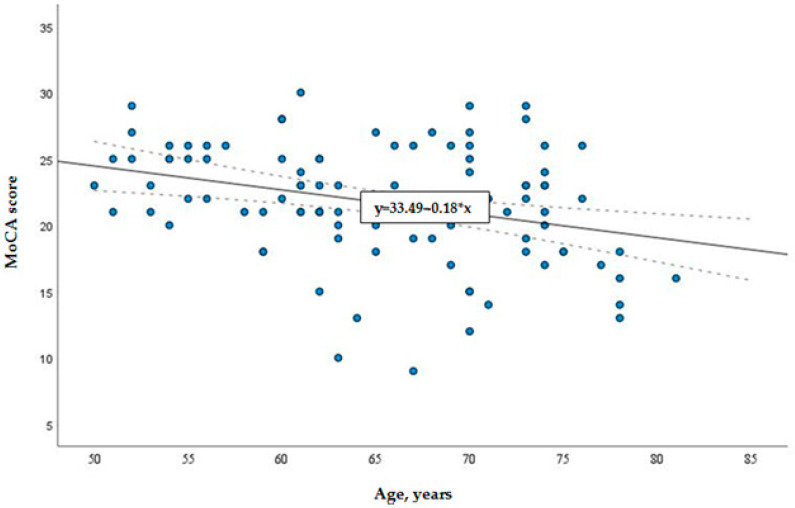
Linear correlation of age with MoCA score.

**Figure 4 medicina-57-00943-f004:**
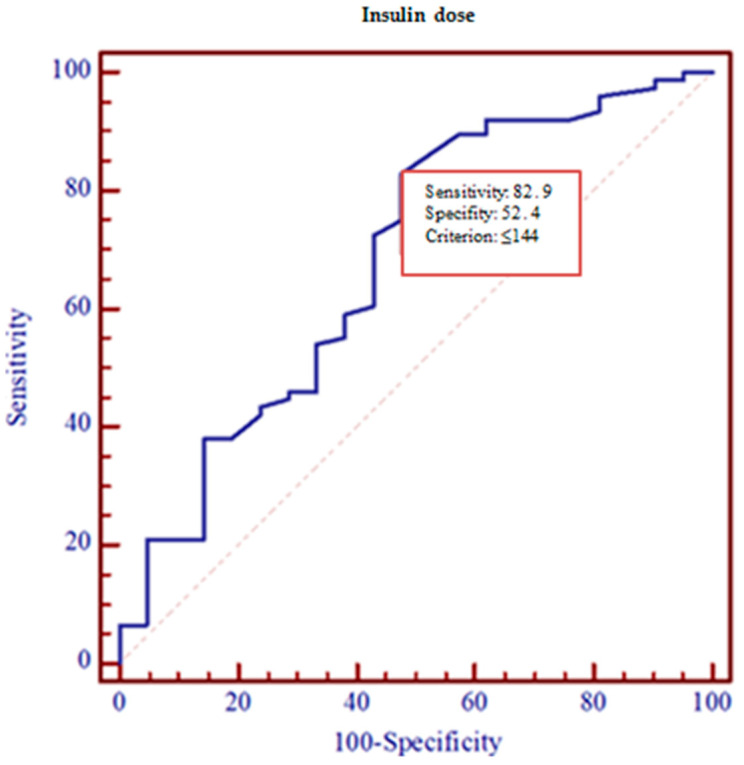
ROC curve for the highest MIC risk.

**Table 1 medicina-57-00943-t001:** Clinical characteristics of participants and the control group.

Characteristics	Case Group	Control Group	*p* Value
Age (years), m ± SD	63.5 ± 7.2	66.5 ± 8.3	0.07
Gender (men/women), %	39.6/60.4	48.1/51.9	0.39
Diabetes duration (years), m ± SD min–max	19.1 ± 8.6	17.2 ± 7.7	0.27
	3–32	5–38	
BMI (kg/m^2^) m ± SD	35.9 ± 6.1	33.9 ± 7.1	0.12
HbA1c %, m ± SD	10.4 ± 1.1	6.9 ± 0.	<0.001
Metabolic syndrome, *n* (%)	44 (84.6)	31 (66.0)	0.03
High (university) education, *n* (%)	15 (28.8)	8 (16.7)	0.14
Marital status: single: *n* (%)	23 (44.2)	18 (37.5)	0.49
Insulin dose per kilo, m ± SD	148.5 ± 36.6	62.7 ± 24.9	<0.001
Hyperuricaemia, *n* (%)	6 (13.0)	11 (25.0)	0.14
MoCA total score, m ± SD	22.2 ± 4.4	21.2 ± 4.4	0.17
Additional metformin use, *n* (%)	18 (37.5)	25 (48.1)	0.56

**Table 2 medicina-57-00943-t002:** Distribution of metformin use according to cognitive functions.

Metformin Use	MoCA Score <26	MoCA Score ≥26	*p* Value
Yes	35	7	0.48
No	40	14	

## Data Availability

We did not report any data.

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
