# Peer review of "Relation between Exogenous Insulin and Cognitive Function in Type 2 Diabetes Mellitus"

_medicina, 2021, doi:10.3390/medicina57090943_

Round 1
Reviewer 1 Report
Authors have made its substantial improvement in respect to previous versions.
However manuscript still requires language editing.
Author Response
we would like to point that our manuscript was checked by a native English-speaking colleague, so moderate English corrections were made.
Reviewer 2 Report
- Which was the minimum type 2 diabetes mellitus duration for enrollment in the study?
- How was the sample size calculated?
- Why did you set an HbA1c < 8% as “sufficient glycemic control”, despite the international recommendations?
- Please specify exclusion criteria. What do you mean with “severe renal impairment” and “active cardiovascular disease”? Please provide time points.
- What about the rest antidiabetic medication? Were novel antidiabetics permitted? Did you address the association of current use of novel antidiabetics with cognitive function (see also the recently published REWIND post-hoc analysis in the Lancet Neurology)?
- Authors should specify the relative frequency of cardiovascular disease in general, coronary artery disease and cerebrovascular disease in their cohort. How many patients in each group received prior treatment with antiplatelet agents, VKAs or DOACs?
- Do the authors propose a diagnostic algorithm for insulin resistant diabetic subjects?
Author Response
Dear Editors and Reviewers,
On behalf of all authors, I would like to thank you for all remarks and questions. We are very glad that first Reviewer has no additional comments after our corrections. We appreciate the second Reviewer for the good insights, which let us to improve our manuscript.
Below, we indicate the responses „point by point“ to the referees’ comments in the attached file.

Round 2
Reviewer 2 Report
I do not have any further comments for the authors.
Author Response
Thank You.
This manuscript is a resubmission of an earlier submission. The following is a list of the peer review reports and author responses from that submission.
Round 1
Reviewer 1 Report
Dear Authors,
my comments are attached.
The authors reported the aim of the study-
- to determine whether insulin resistance, evaluated by daily insulin requirement, has impact on mild cognitive impairment,
- whether this relationship is mediated by insulin doses and other risk factors.
The topic of Type II diabetes and cognitive decline is very important and not enough evaluated, especially with relation to insulin resistance. Therefore, the idea of this study has a scientific value.
And the finding about the highest rate risk of MCI in patients with T2DM and insulin resistance, when insulin dose was less than 144 IU/d - is very important.
However, this manuscript requires a revision.
Below are my comments on this manuscript.
-The title of the manuscript “ Exogenous insulin relation with cognitive function in Type 2 Di-abetes Mellitus in Lithuanian population”. My suggestion – to remove from title “in Lithuanian population”. It is not population study.
-In the Introduction section authors provided an overview of the insulin resistance problem and controversies about the relation to cognitive decline. In the last paragraph of Introduction authors specified the objectives. However- a deeper analysis of the problem, clearer aim of this study and hypothesis are needed.
-In the Introduction section authors discussed the Montreal Cognitive Assessment (MoCA) as a best screening instrument for evaluation of cognitive impairment. My suggestion – to move this paragraph to Methods section.
- my recommendation- to use the same term and abbreviation (Type 2 diabetes mellitus (T2DM)) throughout all the manuscript.
-This study was performed in the Endocrinology department. In methods section would be useful to describe in detail the flow chart of the patients’ inclusion / exclusion: that the study can be reproducible.
-Two different abbreviations were found in the text: MoCA and MOCA. Please unify.
-The sensitivity and specificity of the MoCA questionnaire should be specified.
- MoCA scores below 26 were considered as mild cognitive impairment (MCI). What was the upper level of MoCA for MCI? Whether patients with dementia were excluded?
-To save the space in the manuscript, the results, presented on Fig 1. could be reported in the text or included into Table 1.
-In the Results section the data of correlation analysis was described. My suggestion to prepare a table for a clearer presentation of the results.
-It is not clear the finding: “Participants with lower education tended to have MCI”. Please clarify.
English language editing is suggesting.
Author Response
Please find an attached word file, where You can find Your comments (on the right side of the sheet) and our corrected places (with track changes) .
Thank You once again. The comments were very valuable

Reviewer 2 Report
Dear Authors, I have reviewed your manuscript entitled: Exogenous insulin relation with cognitive function in Type 2 Diabetes Mellitus in Lithuanian population and I have the following comments:
- In the abstract and results Authors should correct significant P-values:
- < 0.05
- < 0.01
- < 0.001
- < 0.0001. Since those values are generally accepted. Thus, in my opinion the P-value p=0.000 should be corrected into < 0.0001 and p=0.00 should be corrected into < 0.001.
- Generally, the manuscript needs some language editing.
- Authors do not need repeat information related to Bioethics committee
- The material and methods sections should be divided into subsections:
2.1. Patient Selection
2.2. Study Design
2.3. Inclusion and Exclusion Criteria
2.4. Methods
2.4.1. Blood Sampling and Laboratory Tests
2.5. Compliance with Ethics Guidelines
2.6. Statistical Methods
- MOCA test should be better described in method section
- Authors, should add precise description of ROC analysis including The area under the curve (AUC) was calculated in order to estimate the diagnostic accuracy. The optimal cut-off points were determined according to Youden criteria.
- General: interesting, well conducted work.
Author Response

(The authors gave the same response as above.)
